# Identifying Novel Emotions and Wellbeing of Horses from Videos Through Unsupervised Learning

**DOI:** 10.3390/s25030859

**Published:** 2025-01-31

**Authors:** Aarya Bhave, Emily Kieson, Alina Hafner, Peter A. Gloor

**Affiliations:** 1Massachusetts Institute of Technology, System Design & Management, Cambridge, MA 02142, USA; bhaveaarya09@gmail.com; 2Equine International, Cambridge CB22 5LD, UK; ekieson@equineintl.org; 3TUM School of Computation, Information and Technology, Technical University of Munich, Arcisstraße 21, 80333 Munich, Germany; alina.hafner@tum.de

**Keywords:** horse emotions, automatic emotion recognition, MoCo, unsupervised learning

## Abstract

This research applies unsupervised learning on a large original dataset of horses in the wild to identify previously unidentified horse emotions. We construct a novel, high-quality, diverse dataset of 3929 images consisting of five wild horse breeds worldwide at different geographical locations. We base our analysis on the seven Panksepp emotions of mammals “Exploring”, “Sadness”, “Playing”, “Rage”, “Fear”, “Affectionate” and “Lust”, along with one additional emotion “Pain” which has been shown to be highly relevant for horses. We apply the contrastive learning framework MoCo (Momentum Contrast for Unsupervised Visual Representation Learning) on our dataset to predict the seven Panksepp emotions and “Pain” using unsupervised learning. We significantly modify the MoCo framework, building a custom downstream classifier network that connects with a frozen CNN encoder that is pretrained using MoCo. Our method allows the encoder network to learn similarities and differences within image groups on its own without labels. The clusters thus formed are indicative of deeper nuances and complexities within a horse’s mood, which can possibly hint towards the existence of novel and complex equine emotions.

## 1. Introduction

In this paper, we demonstrate the inherent ability of convolutional neural network encoders to read horses’ emotions [1] without the requirement for labeled data. Using contrastive learning, we identify well-known emotions of horses automatically, verifying it with manually labelled horse emotion images. Contrastive learning has revolutionized the domain of representation learning through a self-supervised approach that helps eliminate dependence on large labeled datasets. It consists of building up representations that maximize the similarity between positive pairs (semantically similar examples) and the dissimilarity between negative pairs or dissimilar examples so that learned embeddings include discriminative and invariant features. Methods such as SimCLR by Chen et al. [2], MoCo by He et al. [3], and BYOL by Grill et al. [4] demonstrated the popularity of contrastive learning not only in image classification, object detection, and natural language processing but also in many other domains. These methods rely on data augmentations and mechanisms involving memories to gain deeper richness in feature representations.

The same method can be extended naturally to complex problems like emotion detection, where crucial, subtle, context-dependent cues must be captured. For instance, self-supervised models like SimSiam [5] introduced by Chen & He in 2021 and more sophisticated frameworks like SupCon [6] introduced by Khosla et al. in 2020 have been demonstrated to learn robust features even for unbalanced datasets. Contrastive loss-based frameworks, such as [7], have distilled meaningful emotional features from raw audio data and significantly outperformed traditional supervised approaches for emotion detection by speech. More recently, temporal contrastive learning has been found quite effective for video-based emotion recognition, where modeling spatiotemporal dependencies is critical for understanding affective states [8]. These results demonstrate the flexibility of contrastive learning to capture not only static but also dynamic properties of features and make the method an excellent candidate for multi-modal emotion detection tasks. Furthermore, contrastive approaches have proven robust to data scarcity, using augmentations and proxy tasks to generalize across diverse emotional expressions [2,4]. For example, for subtle affective states, the environment is noisy or not controlled; works have used temporal and contextual augmentations to enrich feature representations [8]. Such techniques are applicable in real-world scenarios where labeled data for complex emotions might be limiting or biased. As a result, the potential of contrastive learning can go beyond detecting emotions by covering a promising, yet challenging area that deals with scalable and efficient multi-modal and context-dependent affect recognition.

These advances make contrastive learning particularly well-suited to the application of equine emotion recognition, where expressive signals tend to be subtle and context-dependent. The manifestations of equine emotions include multi-modal cues in the form of facial expressions, body postures, and motion patterns, necessitating a learning framework that incorporates these complexities and interdependencies. Contrastive learning is presented as a promising solution, structuring the feature space in such a way as to illuminate distinctions between emotional states while generalizing across variabilities in expression. This paper attempts to bridge the gap in computational methodologies in equine emotion recognition by providing a pathway toward improved understanding and interaction between humans and animals by using both self-supervised and supervised variants. This also positions contrastive learning as a superior approach to traditional methods for equine emotion recognition, most of which are based on hand-crafted features that are not applicable for capturing the wide scope of emotional variability. The contrastive frameworks draw upon large-scale, unlabeled datasets to discover latent patterns in equine behavior that are hard to model otherwise. Multi-modal data thus used, including visual, motion-based, and contextual cues, provide richer contextual information that could classify equine affective states more holistically. In particular, the temporal contrastive method is capable of modeling sequential dependencies in equine behaviors, which are critical for the identification of emotional state transitions. Such capabilities enhance the robustness of the system while augmenting the interpretability of learned representations. It ultimately focuses on the potentials of contrastive learning as a transformative tool towards further advancements within the domain of animal affective computing with an aspect of somewhat improved welfare and interaction paradigms for human-to-horse relationships.

We have based our research on the seven primal emotions described by Jaak Panksepp. Jaak Panksepp was one of the most influential neuroscientists of our time, and in his research on affective consciousness [9], he suggested that all mammalian species have seven emotional behaviors in common, owing to the fact that all mammals share similar neurological circuitry within their limbic systems. Our primary goal here was to demonstrate that a neural network trained without labels can still differentiate between the seven emotional behaviors. These seven emotions are ‘seeking’, ‘rage’, ‘fear’, ‘lust’, ‘care’, ‘grief’ and ‘play’. In addition to these emotions, we have also included an additional emotional behavior ‘pain’, which aims to identify the subtle cues that indicate that a horse is in pain. For this paper, we have created our own dataset, consisting of 3929 unlabeled horse images. These images were captured by our own equipment and research team, across a variety of wild horse breeds located in different geographical locations. We have used a modified version of Momentum Contrast for Unsupervised Visual Representation Learning (MoCo) [3] framework, developed by Facebook Research in 2019. We have implemented a custom-built downstream classification network. Although there have been previous instances of using MoCo for detecting emotions in other mammals such as dogs [10], unlike dogs, horse emotions are not centralized to their face. A horse’s body posture serves as an indicator of a large proportion of its emotional display. In this paper, we ascertained the effectiveness of previously unused image augmentation techniques and downstream network architectures that optimize posture segmentation and provide an increased accuracy in comparison to traditionally used methods in contrastive learning.

The chief contributions of this paper are:We construct a novel, high-quality, diverse, and unskewed dataset of 3929 images consisting of five wild horse breeds worldwide within different geographical locations. These horse breeds are ‘Eriskay Pony’, found in western Scotland, ‘Insh Konik Pony’, found in the RSPB Insh Marshes, ‘Pottoka’, found in Piornal Spain, ‘Skyros Pony’ found on the island of Skyros in Greece and ‘Konik’ found in the Wicken Fen nature reserve in England. We construct equally sized groups of these images for validation and testing, with labels based around seven Panskepp emotion labels “Exploring”, “Sadness”, “Playing”, “Rage”, “Fear”, “Affectionate” and “Lust”, along with one additional emotion ‘Pain’.We leverage the contrastive learning framework MoCo [3] (Momentum Contrast for Unsupervised Visual Representation Learning) on our dataset to predict the seven Panksepp emotions and ‘Pain’ using unsupervised learning. We significantly modify the MoCo framework to obtain the best possible results on our hardware. We put forth the use of previously unused image augmentation techniques and downstream classification networks that optimize posture segmentation in neural networks trained with contrastive learning.We build a custom downstream classifier network that connects with a frozen CNN encoder that is pretrained using the MoCo [3] framework. This downstream network demonstrates excellent accuracy for its simple architecture and small number of weights.Our method allows the encoder network to learn similarities and differences within image groups on its own without labels. The clusters thus formed are indicative of deeper nuances and complexities within a horse’s emotions, which can possibly hint towards the existence of novel and complex equine emotions.

Our research significantly contributes to the area of Animal–Computer Interaction by providing a novel approach for emotion recognition in horses, promoting improved human–animal communication, high ethical standards, and a contrastive learning approach. The better understanding of horse emotions is thus helpful in filling the human–animal communication gap. Furthermore, we contribute to the field of applied machine learning by demonstrating the efficacy of semi-supervised and self-supervised learning techniques for animal emotion recognition.

## 2. Behavioral Background

### 2.1. Emotions

According to Panksepp (2005), basic emotions in humans and animals can be categorized as Care, Seeking, Fear, Lust, Play, Rage, and Grief [9]. With regard to practical application, all of these basic emotions can be considered spectrums with specific behaviors indicating emotional affect in different animals and ranges of behaviors occurring for each emotion that might indicate the strength of the emotional state. Most of the ethograms and behavioral research on horses fall into categories that do not necessarily align with “emotions”, but rather scientists have studied behaviors that align with physiological indicators of cortisol (indicating stress or irritation) or contextualizing behaviors according to the outcome (such as agonistic, affiliative, or curious). These same researchers have created ethograms to help others better understand these behavioral indicators of emotional affect in horses. Many of these indicators include ear and head position in addition to muzzle tension, body position, leg movement, and tail movement. Ear positions in horses, for example, often help observers determine the “emotion” of the horse based on the correlation of ear position with other behaviors and outcomes, including interactions with humans [11,12], which is further supported by research in cows linking ear position with emotional state [13].

There is little literature showing definitive measures of equine emotional state, a challenge that has been recognized by leaders in the field [14]. Researchers do, however, agree that we can use behavior to better understand emotional states in horses and potentially use these to assess welfare and modify management or handling to improve subjective equine experiences in our domestic environments [14,15,16,17]. There are some categories of emotions for which there is much research (FEAR) and some for which there is little research (CARE). Extensive ethograms exist for areas where there is significant research, while behavioral data from both free-living and feral horses help to provide additional insight into the areas for which there is little research in the domestic world.

While there is limited research on horse emotions, there is research on behaviors and the use of behaviors in horses to communicate desires or express states of being. In order to understand the context of these behaviors with regard to species-specific indicators of emotional valence, environmental variables need to be reduced or eliminated to understand baseline behaviors within a species. Because environments (and variables within domestic settings) can cause variations in behavior, basic behavioral indicators of emotions in horses have been selected from studies where horse behavior has been observed, recorded, and studied in feral and free-living groups. Social behaviors in these groups and behaviors of individuals within groups help serve as the basis for how individual and intraspecies interactions can provide additional insights in behavioral indicators of emotions [18,19,20,21,22,23,24].

#### 2.1.1. Care

Behavioral indicators of the emotion “care” can be generally defined as those that signify nurturing, bonding, affection, or compassion. For horses, these are categorized in the literature as “affiliative” behaviors, which are generally defined as behaviors that are either friendly in nature or that occur exclusively between two individuals who have established a social bond recognized by more time spent in the proximity of one another as opposed to other members of the herd [22,25,26,27,28]. One of the most recognizable affiliative behaviors indicating the emotion of care includes allogrooming, which occurs when two horses stand next to one another, facing in opposite directions, and scratch one another at the same time. The scratching can be at any body part, but usually the horses will scratch each other’s backs or withers (where the neck meets the back) [29,30,31,32,33]. Other behaviors indicating care include moving the head and neck over or under the head or neck of another, the placement of the head on the back of another horse, or when one horse touches their nose against the side of another [27]. In addition to physical interactions, other behaviors indicating care include synchrony of movement (when two or more animals engage in the same activity at the same time) [20,34,35] and shared close proximity for prolonged periods of time [22,23,24,36,37].

#### 2.1.2. Seeking

Behavioral indicators of the emotion for “seeking” include exploration and curiosity behaviors as well as searching behaviors that are expressed when horses are searching for resources and opportunities. Behaviors in this category are aligned with known behaviors of desire, curiosity, exploration, and anticipation. With regard to seeking, there is a specific order of behaviors that indicate curiosity and exploration, starting with looking, then approaching and sniffing, and then often retreating and returning [38].

#### 2.1.3. Fear

Behavioral indicators of the emotion “fear” are associated with behaviors indicating avoidance and panic. Behaviors in this category include everything from mild discomfort and irritation to fear and terror. Much like the other categories, it is a spectrum of behaviors that indicate fear and desire to create more distance between themselves and objects or others or a cautionary approach that results in moving away from something or someone. Most of the behaviors indicating fear have been linked to physiological indicators of stress, mostly cortisol, that have been taken from blood, saliva, feces, and hair (indicating long-term stress responses) [39,40,41,42,43,44,45,45]. Behaviors in this category usually include high head position, wide eyes (where the white is visible), and fast or slow movement away from a fearful object [46,47].

#### 2.1.4. Lust

Behavioral indicators of the emotion “lust” in horses can be defined as those related to sexual and reproductive behaviors or behaviors indicating a desire to lead to reproductive behaviors. Reproduction and related behaviors in horses have been studied extensively for years as part of efforts to make reproduction more efficient in the domestic world. In horses these behaviors vary between mares (reproductive females) and stallions (reproductive males) and range from mares lifting their tails and “winking” their vulvas to stallions biting the mare and mounting [48,49,50,51].

#### 2.1.5. Play

Play has been studied extensively in horses and, as an emotional category, looks at behaviors indicating joy and social engagement through active, fun bonding. There are two general categories of play: object play and play between conspecifics [52]. Since these behaviors often momentarily mimic rage, many behaviors categorized as play may also be similar to those that indicate rage and fighting. Ear positions can help differentiate between play and rage, as can level of arousal (stress) and ability to regulate arousal. Common behaviors indicating play include biting each other’s legs, neck or mouth with light or little contact, momentary pinning of the ears (when ears are flattened back against the neck), slow running or sparring, and ability to regulate movements (slow upwards or downwards progression of movement with no sudden bursts and no fear responses). Play often mimics fighting (in the emotional category of rage) but does not include any hard contact, and both animals continue to choose to remain in the proximity of the other even after contact play has ended.

#### 2.1.6. Rage

Behavioral indicators of the emotion “rage” can be defined as those that signify frustration, anger, or irritation. Behaviors indicating rage take place when one individual feels this kind of emotion towards another living creature or towards an environmental condition or an interaction (including with people and riders). Interspecific communication of emotions is usually towards a conspecific, although horses will display these communication behaviors towards other animals, including humans, or in domestic conditions where they are communicating a mild to extreme physical or psychological discomfort [53,54]. For horses, the emotion of “Rage”, like other emotions, is a spectrum ranging from irritation or dislike to full anger and rage and can be broken into two categories: Intra- and Interspecies communication. With regard to intra-species rage, we can look at pro-social behaviors that occur between equine conspecifics in the category of agonistic behaviors that indicate when one horse does not like the physical presence of another horse. In general, these behaviors are short in duration and last until the desired space has been achieved. Like stress behaviors, agonistic behaviors have been well studied under different circumstances with full ethograms describing the behavioral indicators [55,56]. These behaviors include pinning of the ears back against the neck, wide eyes, a tight muzzle, bite threats, biting, kick threats, kicking, tail swishing, head tossing, and striking out.

#### 2.1.7. Grief

Behaviors aligned with emotions of grief are considered indicators of sadness, social distress, or social loss. Along with sadness, we can also look at potential signs of depression in this category. Although depression is considered a human diagnosis, researchers have looked at ethological animal models of depression with a focus on horses [57]. These behaviors include lack of movement or motivation to move, lack of curiosity or seeking behaviors, loss of appetite, and behavioral indicators that are also in line with mild physical discomfort, such as chronic lowering of the head and loss of interest in social interactions.

### 2.2. Use of Emotions in Research

Scientists suggest that indicators of emotional states, including behaviors aligning with positive emotional wellbeing, should be included in assessments of psychological wellbeing and welfare in horses. Researchers have also assessed which of these behaviors they can use to assess equine welfare and wellbeing in the domestic setting.

Mutual grooming, also known as allogrooming, occurs when two individuals engage in scratching one another with their upper lip or teeth at the same time, usually at the withers or back, and is considered one of the behavioral indicators of the emotional category of care. In feral horses, this is seen only between two individuals who have demonstrated spatial preference and have shown signs of social bonding through time and proximity [25]. This kind of known affiliative interaction is often more frequently displayed in domestic horse herds than in feral herds [58,59], suggesting that increased frequency of allogrooming and play are often linked with coping behaviors associated with chronic or acute stress [31,60,61]. Recently, researchers have suggested that this correlation of high frequency of affiliative behaviors with stress means that these behaviors are not good indicators of positive welfare in domestic herds of horses and that additional indicators should be used to assess domestic horse welfare [28].

To control for unnatural physical and social settings that might influence the type, frequency, or behavioral expression of emotions, it is critical to base these kinds of behavioral studies on free-living and feral horses where horses have choice and freedom to express behaviors based on natural social and environmental conditions. Such studies of free-living, feral, and wild horses serve as the basis for understanding social behaviors in horses to control for the variables of the domestic setting and to provide additional insights into the natural lives of horses [21,24,62]. Data used for this study have therefore focused on videos of free-living and wild horses to avoid potential influences of the constructed domestic world on the emotional expression of horses.

If we can accurately measure or categorize behaviors in horses, we can therefore take steps to manage conditions that either favor positive emotional states or decrease negative ones [17].

## 3. Related Work

Recent advancements in animal motion tracking and pose recognition have significantly impacted the study of animal behavior, enabling automated recognition of internal states such as emotions and pain to improve animal welfare. Broomé et al. [63] present a comprehensive survey of computer vision-based research on recognizing pain and emotional states in animals, analyzing both facial and bodily behaviors.

Additional recent research has been done to assess the use of machine learning in recognition of horse expressions and behavior related to welfare, specifically in using the equine pain face [64] and grimace scale [65,66]. Studies in 2018 and 2021 supported the use of machine learning to recognize facial expressions of horses, especially related to the research on facial expressions related to pain [67,68]. Researchers have used this preliminary research to specifically assess the ability of automated systems to recognize emotional expressions in horse faces [69] and develop deep-learning models to further read pain expressions in equine faces [70] by using AI models to recognize horse emotions from facial expressions in a controlled experiment. They tested two approaches: A deep learning pipeline using video footage and a machine learning pipeline utilizing EquiFACS annotations. The deep learning model outperformed, achieving 76% accuracy in distinguishing four emotional states—baseline, positive anticipation, disappointment, and frustration. However, separating anticipation and frustration proved challenging, with only 61% accuracy. Machine learning has expanded beyond horse to recognize facial expressions in other animals [71]. Additional research in artificial intelligence and machine learning has also supported the use of these technologies to automate behavioral recognition with guided training by specialized animal behaviorists [72]. The research in using machine learning to read and recognize facial expressions in horses is growing, but there is a lack of work in using machine learning to recognize and assess the emotional expressions of horses using body position or movement.

Tendencies towards using unsupervised methods (e.g., [73]) can be identified as offering promising results. With our work, we extend previous research that mainly employed supervised ML methods for detecting emotions and well-being in horses. In contrast to using video data, previous research mainly uses image data. For example, Corujo et al. [74] designed a “proof of concept” system to recognize emotions in horses, combining a fast region-based convolutional neural network (detector) for identifying horses in images and a convolutional neural network (model) for predicting their emotions. The system was trained on 400 labeled images and tested on 40 images, achieving an 80% accuracy on the validation set and 65% on the test set. Four emotional states (alarmed, annoyed, curious, and relaxed) were inferred using behavioral ethograms, focusing on head, neck, ear, muzzle, and eye positions.

With respect to unsupervised emotion detection, Bhave et al. [10] demonstrate how novel canine emotions can be detected using the Momentum Contrast (MoCo) framework. They developed a system to identify dog emotions based on facial expressions and body posture. To do so, they constructed a dataset comprising 2.184 images of ten popular dog breeds, categorized into seven primal mammalian emotions as defined by Jaak Panksepp: Exploring, Sadness, Playing, Rage, Fear, Affectionate, and Lust. Using a modified version of the contrastive learning framework MoCo, they achieved an unsupervised accuracy of 43.2% on their dataset, significantly surpassing the 14% baseline. When applied to a second public dataset, the model achieved 48.46% accuracy against a 25% baseline. This research shows that an unsupervised approach can achieve promising results.

## 4. Method

In this experimental study we have pursued a novel approach that focuses on giving a convolutional neural network freedom to learn on its own, reconfiguring the image augmentation layers and framework architecture. Consequently, we have prepared an original dataset of unlabeled images. We have developed a pipeline that we used to extract high-quality horse images from over 3 terabytes of original horse video footage. To conduct a comprehensive representation learning experiment on these data, we used the MoCo [3] framework. The decision to use this framework was inspired by previous work on dog emotions [10], where a detailed study on contrastive learning frameworks such as SimCLR [2] was conducted to determine the best performing framework for dog images. Motivated by the previous successful application of the MoCo framework and its low computing infrastructure requirements, for this study we have implemented a modified version of the MoCo framework [3] developed by He et al.

### 4.1. Creating the Dataset

As mentioned previously, we had access to over 3 terabytes of proprietary video footage. This footage was collected exclusively from wild and untamed horses to preserve the natural element within each emotional display. Our team traveled to five different locations to obtain the footage of five different wild horse breeds. Table 1 describes the different wild horse breeds from which our data were collected. To ensure the diversity and representativeness of the dataset across different horse breeds and geographical locations, we selected breeds that have distinct ancestral backgrounds and morphological characteristics. Each horse breed is associated with different conformational traits, distinct sizes (e.g., the smaller stature of the Skyros ponies vs. the larger Koniks), and coat colors. This allows for a broader comparison of phenotypic variation within and across semi-feral and wild horse populations.

#### 4.1.1. Dataset Description

We created an evenly distributed dataset, collecting a similar number of data points from all five aforementioned horse breeds. Figure 1 gives the distribution of image count across the five breeds. There are 711 Eriskay Pony images, 781 Insh Konik images, 809 Pottoka images, 796 Skyros Pony images and 832 Konik images.

#### 4.1.2. Splitting the Dataset

In order to conduct contrastive pre-training and then fine-tune a downstream classifier network after freezing the pre-trained encoder, we needed to split the dataset into three sections. The three sections are each individually required for contrastive pre-training, downstream fine-tuning and final testing, respectively. These splits were conducted in a stratified shuffle in order to maintain the same breed distribution in each of these splits. We partitioned 86% of our data, amounting to 3374 images for contrastive pre-training using the MoCo [3] framework. We partitioned 10% of our data, amounting to 396 images for fine-tuning the downstream classifier network. Finally, the remaining 4% of the data, i.e., 159 images, were used to test the performance of the final model.

Figure 2 describes the split ratio in the stratified-shuffle split. Figure 3 and  Figure 4 describe the emotion distribution in the 10% and 4% splits, respectively. It is important to note that our team classified 14% of the total collected data, amounting to 555 images, into seven Panksepp [9] and the ‘Pain’ emotions manually. Ashley et al. described a comprehensive method of deciphering the ‘Pain’ emotion among equines in [75]. Kieson et al. provide a unambiguous idea of identifying affection and lust among equines in [27,31]. McDonnell et al. describe the playful behavior in horses in [52]. Burke explains in detail the movement patterns associated with exploratory and seeking behaviour among horses in [38]. Fureix et al. give an insight to the display of grief in horses in [57]. Leiner gives a detailed analysis of fear and rage among horses in [76]. Our team has adhered to the directives established by the above research papers while labeling the few hundred images for downstream tasks and testing. It is important to note that labeled data are only required to analyze the performance of the encoder that was trained using unlabeled data.

### 4.2. Image Retrieval Pipeline

With over 3 terabytes of video footage, collected in different devices across different locations, manually extracting horse images would have been a futile effort. Each frame may contain multiple horses and since a significant proportion of footage was collected without a tripod, some frames may have a motion blur. As a result, we resorted to developing an automated system, powered by YOLOv8 object detector [77], which would automatically identify the horses. Following this, we set up an image filter phase, that made sure any images with extremely low resolution and high noise were eliminated. We maintained a minimum resolution of 250×250. We also maintained a minimum required value of pixel intensity standard deviation ${œ}_{x}$ after normalizing pixel values, such that ${œ}_{x}<0.4$.

Figure 5 describes the basic flow of data within our image retrieval pipeline. We used this pipeline on all the videos our team had collected, and through this pipeline we were able to extract the best cropped horse images to constitute a dataset. To achieve this, frames are taken from the video, initially at 30 FPS but later downsampled to 0.1 FPS for more efficient processing. To do this, the heaviest version of the object detection model ‘YOLOv8x’ [77], is adopted because it is highly accurate and efficient in recognizing objects like horses and generating a bounding box around them in those frames. The detected bounding boxes are cropped, and their images are filtered for standard deviation of pixel intensity and resolution. Finally, the filtered cropped horse image are stored in a dataset for further augmentation and training.

The pipeline starts with a high-frame-rate video at 30 FPS, which is then reduced to 0.1 FPS to prevent redundant frame processing. YOLOv8x [77] detects horses in the frames, using its 68.2M parameters and 257.8 GFLOPs computational power. Each detection is represented as a bounding box around the identified horses within the frames. During crop extraction, only those bounding box regions with low noise $\left({œ}_{x}<0.4\right)$ and sufficient resolution are maintained for high-quality data. These are meant for use in tasks such as training ML models or for analysis within a domain for the purpose of analyzing horse images.

Using this pipeline, we were able to collect 3929 high-quality horse images that we used for our project. Contrastive Learning is a data-intensive learning technique, and thus we reserved 3374 unlabeled images for pre-training the encoder.

### 4.3. Momentum Contrast for Unsupervised Visual Representation Learning

Contrastive learning has become a prominent paradigm in self-supervised learning, where it draws upon the intuition of bringing similar representations closer and pushing dissimilar ones apart in a latent space as seen in the research work of Hadsell et al. [78] and Chen et al. [2]. It is effective because it learns strong features that generalize across tasks by maximizing mutual information between augmented views of the same data, as explored by Oord et al. in [79]. Momentum Contrast [3] by He, Yang, Craswell, Wang, Chen, and Gao in 2020 advances this framework by providing a dynamic dictionary with a momentum encoder that allows for large-scale instance discrimination using a memory bank to retain a representation space. Unlike other frameworks, such as SimCLR by Chen et al. [2], 2020, which requires large batch sizes and intense computation, or BYOL by Grill et al. [4], 2020, which abandons negative samples, MoCo balances between the decoupling of batch size and dictionary size, ensuring that negatives are always of high quality by using a momentum mechanism. Its flexibility and scalability have made it a cornerstone in unsupervised representation learning, bridging the gap between self-supervised and supervised performance in vision tasks.

#### 4.3.1. Data Augmentation Layer of MoCo

The data augmentation layer of the Momentum Contrast (MoCo) framework is the core component of generating diverse and meaningful views of the input data for enhancing the representation learning process. Inspired by prior works like SimCLR [2], MoCo uses strong augmentations, such as random cropping, resizing, horizontal flipping, color jittering, grayscale conversion, and Gaussian blurring, to create two augmented versions of the same image. Such augmentation ensures learning invariant features that survive distortion and transformation as envisioned by contrastive learning objectives, such as those introduced by Hadsell et al. [78]. The augmentation pipeline of MoCo prevents the quality degradation of positive pairs (of the same instance), which in turn allows contrasting from the negative pairs (whose instances are different) by residing in the memory bank for effective discriminative representations. By spectrum augmentation of input data to variation without losing any semantic content, the MoCo augmentation layer is a critical component.

The primary function of the augmentation layer is to emphasize the invariant features within an image. This implies that an augmentation layer should be kept flexible, to see which components within the layer perform best. The original MoCo [3] augmentations were borrowed from SimCLR [2], which itself was trained on the ImageNet [80] dataset. Specifically, SimCLR [2] was trained on the ILSVRC-2012 version, which contains over 1.2 million labeled images across 1000 categories. Our dataset contained only 3374 images for training. Learning from [27,31,38,57,75,76], we discovered that a horse’s posture and the relative position of its neck, head, and tail are the most significant indicators of its emotions. As a result, we developed an augmentation layer with serially organized components that are designed to accentuate these features within a horse image. At the same time, we made sure that our augmentations got rid of any irrelevant features, like the horse’s color and the surrounding terrain. Consequently, we independently developed the query transforms and the key transforms.

This pipeline is carefully designed to introduce randomness in such a way that it keeps the input data structurally intact to enable contrastive learning. The **query_transforms** pipeline initiates with a random resized cropping operation parameterized with a scaling range of (0.66,1.0), thus it diversifies the spatial representation of a particular image by extracting variable-sized subregions from the original image. The subsequent grayscale conversion removes the color information and allows the model to work on intensity and texture pattern variations. A random horizontal flip with a 0.5 probability introduces controlled geometric distortion and enhances orientation insensitivity. Finally, Gaussian blur, used sometimes (with a 0.1 probability), is also an injection noise method that emulates realistic distortions and hence enhances the robustness of learned features. The Sobel-Filter distinctly highlights gradient information, facilitating accurate extraction of features based on edges. Ultimately, the stages of tensorization and normalization establish compatibility with GPU-accelerated computations while ensuring uniform input distributions, thereby promoting effective optimization.

The **key_transforms** pipeline is a complementary augmentation strategy that has a focus on structural integrity, with a lower degree of stochastic noise. It starts with a random resized crop with a constrained scale range of (0.85,1.0), so that more contextual spatial information is likely preserved in the image. It is similar to the query branch and applies grayscale conversion as well as the Sobel-Filter for edge and texture preprocessing necessary to maintain feature consistency across different views. This pipeline deliberately excluded horizontal flips and blur augmentation in order to allow a more deterministic baseline representation aligned with the broader strategy of multi-view contrastive learning. The transformation pipeline finally ends with tensor conversion and normalization for ensuring consistent comparability across inputs from any of the augmentation streams. Pipelines work in concert within the TwoCropsTransform framework to produce contrasting yet complementary views that facilitate the learning of robust, invariant feature embeddings, which are critical for unsupervised representation learning.

The following Listing 1 describes our data augmentation layer in terms of query and key transforms:

Figure 6 displays the work of the image augmentation layer.

**Listing 1.** Query and Key Transforms.

1 query_transforms = transforms.Compose([

2     transforms.RandomResizedCrop(size=(SIZE, SIZE), scale=(0.66, 1.0)),

3     transforms.Grayscale(),

4     transforms.RandomHorizontalFlip(p=0.5),

5     transforms.RandomApply([GaussianBlur()], p=0.1),

6     SobelFilter(),

7     transforms.ToTensor(),

8     normalize

9 ])

10  

11 key_transforms = transforms.Compose([

12     transforms.RandomResizedCrop(size=(SIZE, SIZE), scale=(0.85, 1.0)),

13     transforms.Grayscale(),

14     SobelFilter(),

15     transforms.ToTensor(),

16     normalize

17 ])

18 train_transforms = TwoCropsTransform(query_transforms, key_transforms)


#### 4.3.2. Encoder

The augmentation layer produces images in a single channel, as it is designed to convert all images to grayscale. As a result the traditional encoders used with MoCo are not the optimal choice here. The MoCo paper discussed experimentation with ResNet-50-w4x, ResNet-50w2x and ResNet-50 as encoders. These encoders are not only resource-intensive but also difficult to train with limited hardware. The most significant issue with using these encoders is that they require a large amount of GPU video memory (VRAM). The primary reason behind this is that these encoders are designed to handle data inputs with 3 channels (RGB). By grayscaling our inputs, we not only got rid of the irrelevant color data but also made it easy to train with limited VRAM and computing resources. Consequently, we decided to utilize a completely new encoder architecture, inspired by SOPCNN [81]. With 1,400,000 trainable parameters, MNIST-SOPCNN [82] is ranked 4th in the accuracy rankings for MNIST Image Classification [83], with an accuracy of 99.83%. SOPCNN, standing for Stochastic Optimization Plain Convolutional Neural Networks is an optimization technique that uses a combination of regularization techniques that work together to get better performance, making plain CNNs less oversensitive to overfitting. Our SOPCNN-inspired encoder is a plain CNN structure dedicated to efficient image classification, based on the tenets of simplicity and efficiency as found in [81]. This model systematically follows the implementation of four convolutional layers characterized by ReLU activations with the use of 3 × 3 kernels and 2 × 2 max-pooling for reducing spatial dimensions while extracting hierarchical features simultaneously. The architectural design ends with a densely connected layer with 2048 neurons followed by dropout regularization with a parameter *p* = 0.8 to prevent overfitting by randomly disabling neurons in the training process. It is designed for grayscale input and allows images of different sizes, which is consistent with leading performance on datasets like MNIST and SVHN. Thus, it shows the efficiency of conventional CNNs if appropriately enhanced through robust regularization methods like dropout and data augmentation. It is important to note that MoCo [3] uses identical architectures in both query and key encoders.

The primary reason for selecting the SOPCNN encoder over traditional encoders like ResNet-50 is that it is better suited to handle output produced by our custom Data Augmentation Layer. It efficiently operates on a single channel input and is much better at segmenting the features accentuated by the Sobel Filters present in the data augmentation layer. In addition to that, we also require lesser computing resources like VRAM when working with SOPCNN.

The following Listing 2 describes our query and key encoder:

**Listing 2.** Query and Key Encoder.

1 class SOPCNN(nn.Module):

2     def __init__(self, num_classes=128, IMG_SIZE=224):

3         super(SOPCNN, self).__init__()

4         self.IMG_SIZE = IMG_SIZE

5         self.conv1 = nn.Conv2d(1, 32, kernel_size=3, padding=1)

6         self.conv2 = nn.Conv2d(32, 64, kernel_size=3, padding=1)

7         self.conv3 = nn.Conv2d(64, 128, kernel_size=3, padding=1)

8         self.conv4 = nn.Conv2d(128, 256, kernel_size=3, padding=1)

9         self.pool = nn.MaxPool2d(kernel_size=2, stride=2)

10         self.fc1 = nn.Linear(256 * (self.IMG_SIZE//4) * (self.IMG_SIZE//4), 2048)

11         self.fc2 = nn.Linear(2048, num_classes)

12         self.dropout = nn.Dropout(0.8)

13     def forward(self, x):

15         x = F.relu(self.conv1(x))

16         x = F.relu(self.conv2(x))

17         x = F.relu(self.conv3(x))

18         x = F.relu(self.conv4(x))

19         x = self.pool(x)

20         x = x.view(-1, 256 * (self.IMG_SIZE//4) * (self.IMG_SIZE//4))

21         x = F.relu(self.fc1(x))

22         x = self.dropout(x)

23         x = self.fc2(x)

24         return x


The proposed architecture, SOPCNN, is a custom convolutional neural network designed for feature extraction and classification tasks. The network processes single-channel images of size 224×224 by default but can be configured for other dimensions by adjusting the IMG_SIZE parameter. The architecture consists of the following components:**Convolutional Layers:** The network includes four convolutional layers:-The first layer (conv1) applies 32 filters of size 3×3 with a padding of 1 to preserve spatial dimensions.-The second layer (conv2) increases the feature map depth to 64, with the same kernel size and padding.-The third layer (conv3) expands the depth to 128, maintaining the kernel size and padding.-The fourth layer (conv4) extends the depth further to 256, also maintaining the kernel size and padding.Rectified Linear Unit (ReLU) activation is applied after each convolutional layer.**Pooling Layers:** Two max-pooling layers, each with a kernel size of 2×2 and a stride of 2, reduce the spatial dimensions of the feature maps. These pooling layers are applied after the second and fourth convolutional layers, halving the image dimensions at each stage.**Fully Connected Layers:**-The first fully connected layer (fc1) transforms the flattened feature map into a 2048-dimensional vector.-The second fully connected layer (fc2) reduces this vector to the desired number of output classes, specified by the num_classes parameter (default: 128).**Dropout:** A dropout layer with a probability of 0.8 is included after the first fully connected layer to prevent overfitting.**Activation and Output:** ReLU activation is applied after all convolutions and the first fully connected layer, while the final layer outputs logits for classification tasks.

The forward pass of the network involves sequentially applying the aforementioned operations, with the feature maps being flattened appropriately before entering the fully connected layers. This architecture is designed to balance computational efficiency and feature extraction capability for a wide range of classification tasks.

### 4.4. Final MoCo Network Architecture

Figure 7 displays the work of the image augmentation layer.

The SOPCNN Query Encoder is then trained for sufficient rounds and its settings are frozen. Another network is then attached to its output to see how well it performs. In this phase, the progress is tracked with a loss value that decreases. The NT-Xent (Normalized Temperature-Scaled Cross-Entropy) loss is derived as follows:

Let the embeddings $z_{i},z_{j}\in R^{d}$ be normalized to unit norm, i.e., $∥z_{i}∥=1$ and $∥z_{j}∥=1$. The cosine similarity between two embeddings is defined as:(1)$$\mathrm{sim}\left(z_{i},z_{j}\right)=z_{i}\cdot z_{j}.$$
To control the sharpness of similarity scores, a temperature parameter τ>0 is introduced, scaling the similarity as:(2)$$\mathrm{scaled}\text{\_}\mathrm{sim}\left(z_{i},z_{j}\right)=\frac{\mathrm{sim}(z_{i},z_{j})}{\tau }.$$
Using this scaled similarity, we compute the exponential similarity (logits) for a pair (i,j):(3)$${\mathrm{logit}}_{ij}=\exp\left(\frac{\mathrm{sim}(z_{i},z_{j})}{\tau }\right).$$
For a positive pair (i,j), the probability $p_{ij}$ is defined using a softmax function, normalized over all samples in the batch except *i*:(4)$$p_{ij}=\frac{{\mathrm{logit}}_{ij}}{\sum_{k=1}^{2N}1_{\left[k\neq i\right]}{\mathrm{logit}}_{ik}},$$
where $1_{\left[k\neq i\right]}$ is an indicator function ensuring *i* is excluded from the denominator.

Substituting Equation (3) into Equation (4), the probability becomes:(5)$$p_{ij}=\frac{\exp\left(\frac{\mathrm{sim}(z_{i},z_{j})}{\tau }\right)}{\sum_{k=1}^{2N}1_{\left[k\neq i\right]}\exp\left(\frac{\mathrm{sim}(z_{i},z_{k})}{\tau }\right)}.$$
The loss for a positive pair (i,j) is the negative log-probability:(6)$$\ell_{i,j}=-\log p_{ij}.$$
Substituting $p_{ij}$ from Equation (5) into Equation (6), we get:(7)$$\ell_{i,j}=-\log\frac{\exp\left(\frac{\mathrm{sim}(z_{i},z_{j})}{\tau }\right)}{\sum_{k=1}^{2N}1_{\left[k\neq i\right]}\exp\left(\frac{\mathrm{sim}(z_{i},z_{k})}{\tau }\right)}.$$

With NT-Xent Loss function stochastic gradient descent, the query encoder is improved. At the same time, the momentum encoder is updated using exponential moving averages. According to the method from MoCo [3], the encoder is designed to produce the SOPCNN output embedding vectors of size 64 for downstream usage.

## 5. Results

Throughout this project, we have made extensive use of PyTorch, which is a deep learning and hardware acceleration library in Python v.3.9. While experimenting locally with our NVIDIA RTX 3070-Ti GPU, we have used Ubuntu-22.04 LTS, which is an opensource Linux-based operating system built by Canonical, London, England. While training on the NVIDIA Tesla P100 GPU, we have used Kaggle’s default shell for operating the Linux commands during training and analysis. Locally, we have used an Intel-12800HX CPU, manufactured by Intel, Santa Clara, CA, United States, and in Kaggle’s remote environment we have used their Intel-Xeon CPU. It is important to note, however, that the CPU makes little difference for the outcome of our experiments.

### 5.1. Pre-Training with Momentum Contrast

As discussed previously in the manuscript, we have designed the encoder ourselves, taking inspiration from SOPCNN [81], while training this encoder using momentum contrast [3], we used the following hardware shown in Table 2:

Using the NT-Xent Loss function discussed earlier, we were able to train the encoder and track the training progress. Owing to the novelty of this encoder, we had to work through different sets of hyper-parameters and after thorough experimentation, we found the following hyperparameter values to be the most optimal, shown in Table 3. Our method of determining optimal hyperparameters was trial and error, where we manually adjusted the hyperparameter values according to the trends we discovered throughout the training process. It is important to note, however, that the hyperparameters ‘Batch Size’, ‘Input Size’, ‘Queue Capacity’ and ‘Representation Dimension’ in Table 3 may be sub-optimal for other machines, since we were largely limited by our hardware. With the use of an industry-grade data center that has lots of VRAM, it is possible to obtain better results in this experiment.

After 2000 epochs spent in training, we observed the following decrease in loss value Figure 8:

In order to track the training progress of our encoder, after every 100 epochs, we analyzed the features produced by a single forward pass of the encoder. This was done by taking the output vector embeddings from the encoder’s output, and using t-distributed Stochastic Neighbor Embedding (t-SNE) to reduce its dimensions from 64D to 2D. The perplexity value we used for this was 30. Figure 9 shows the distribution of the 3374 datapoints before training the encoder. Figure 10 shows the distribution of these datapoints after 2000 epochs of training with the aforementioned hyperparameters.

We can visibly note the cluster formation that has occurred, indicating that the model was inherently able to distinguish between the emotions in an unsupervised fashion. Further visual analysis indicates that the encoder clustered these data into more than 8 clusters. This means that the encoder has clearly picked up on more than the eight original emotions. This study was originally developed around the seven Panksepp emotions [9] and the ‘Pain’ emotion, but this result foreshadows the possibility of the existence of novel emotions among equines. We hypothesize that our SOPCNN encoder has detected novel emotions that are more complex and nuanced and that academia may not have been aware of until now.

### 5.2. Downstreaming

In this section, we explore the performance of our encoder, by systematically freezing its weights and linking it to a downstream classifier as follows in Listing 3:

**Listing 3.** Downstream Architecture.

1 class CompleteModel(nn.Module):

2     def __init__(self, encoder, feature_dim=64, num_classes=8):

3         super(CompleteModel, self).__init__()

4         self.encoder = encoder

5         for param in self.encoder.parameters():

6             param.requires_grad = False

7         self.fc1 = nn.Linear(feature_dim, 256)

8         self.bn1 = nn.BatchNorm1d(256)

9         self.dropout1 = nn.Dropout(0.5)

10         self.fc2 = nn.Linear(256, 128)

11         self.bn2 = nn.BatchNorm1d(128)

12         self.dropout2 = nn.Dropout(0.5)

13         self.fc3 = nn.Linear(128, 64)

14         self.bn3 = nn.BatchNorm1d(64)

15         self.dropout3 = nn.Dropout(0.3)

16         self.fc_out = nn.Linear(64, num_classes)

17  

18     def forward(self, x):

19         features = self.encoder(x)

20         x = F.relu(self.bn1(self.fc1(features)))

21         x = self.dropout1(x)

22         x = F.relu(self.bn2(self.fc2(x)))

23         x = self.dropout2(x)

24         x = F.relu(self.bn3(self.fc3(x)))

25         x = self.dropout3(x)

26         x = self.fc_out(x)

27         return x


After thorough experimentation, we discovered that the above downstream architecture is superior for accurate classification. The proposed network architecture consists of a **pretrained SOPCNN encoder** combined with a **custom classification head** designed for efficient transfer learning and robust feature refinement. The pre-trained encoder acts as a feature extractor; its parameters are frozen to retain the learned representations from a large-scale source dataset. This reduces the training time and computational resources. The downstream classifier comprises the following layers:(1)**Fully Connected Layers (fc1, fc2, fc3):** These layers progressively reduce the feature dimensions from the encoder output (feature_dim = 64) to align with the classification task. This dimensionality reduction enables the network to extract higher-order representations tailored to the downstream task.(2)**Batch Normalization (bn1, bn2, bn3):** Batch normalization is applied after each fully connected layer to stabilize learning by normalizing feature distributions. This helps accelerate convergence and mitigates internal covariate shift.(3)**ReLU Activations:** Using the activation function ReLU in the hidden layers introduces a nonlinearity that the network then uses to approximate complex interrelations in data.(4)**Dropout (dropout1, dropout2, dropout3):** Utilize dropout to prevent the model from overfitting by randomly disabling neurons by using different dropouts; dropout1 0.5 and dropout2 0.3. The model learns patterns that could be generalized.(5)**Output Layer (fc_out):** A final, fully connected layer maps the refined features to the required number of classes (num_classes = 8), serving as the prediction layer for classification.

We trained this downstream classifier, with the aforementioned fine-tuning and validation datasets, which each contain 396 and 156 labeled images, respectively. After training the above architecture for 500 epochs, we found that the following trend emerged Figure 11:

The classifier was able to achieve a classification accuracy of 55.47% across 8 labels, where the baseline accuracy would be 12.5%. We observed the following accuracy scores while evaluating individual emotions. We ascertain that the variations in accuracy values for different emotions are primarily caused because of the subtlety and prominence of each emotional display. Some emotions like ‘Lust’ and ‘Pain’ are displayed very prominently by the horse, while other emotional displays like ‘Seeking’ and ‘Grief’ are much less pronounced in a horse’s body and posture. Some of these emotions may require a series of consecutive images to confirm, which opens up the aspect of using temporal data. See Table 4.

### 5.3. Generalizability of Our Results

Our results demonstrate the strong potential of contrastive learning frameworks, particularly the MoCo variants, for analyzing emotional behaviors in horses. The effectiveness of our models, despite using lighter encoders and operating within computational constraints, underscores the adaptability and robustness of these approaches. However, it is essential to recognize the limitations posed by the specific dataset and the potential variability in real-world scenarios. Future research should focus on evaluating these models across diverse datasets, including those representing different breeds and environments, to ensure broader applicability. In addition, techniques of transfer learning may improve the generalizability, such that models trained on horse emotion datasets can be applied to other species or even human emotional recognition tasks. This cross-species application may open new avenues for understanding and interpreting emotional behaviors using machine learning, thus fostering advancements in both animal behavior research and human–animal interaction studies.

Our findings hold significant potential for broader application. Successfully generalizing these models to diverse datasets could lead to a deeper and more accurate understanding of horse emotions in various settings, such as among different breeds or environments. This progress could drive innovations in fields like animal welfare, training, and therapy. Furthermore, utilizing transfer learning methods may allow these models to be adapted for analyzing emotions in other species or even humans. Such advancements could contribute to a broader comprehension of emotional behavior, enhancing research on human–animal interactions and improving emotional response strategies across species.

Our current model relies solely on postures, which may limit its accuracy in distinguishing between multiple emotions. Equine emotions are influenced not only by postures but also by environmental and social factors, such as the presence of other horses, animals, or humans. To address this limitation, future experiments should focus on incorporating contextual information by training on rich, diverse datasets that capture a wide range of interactions. This can be achieved through multimodal approaches or context-conditioned training, enabling the model to better interpret and differentiate subtle emotional variations in horses.

### 5.4. Comparative Analysis with Other Relevant Methods

In order to understand the full scope of this research and this method of representation learning, we must view it comparatively. The closest and the most relevant study that can be used to establish a comprehensive comparative analysis is our previous work with canine emotions. There are several parallels between this study and our previous work [10] titled ‘Unsupervised Canine Emotion Recognition Using Momentum Contrast’, which will allow us to ensure the fairness and rationality of the experimental study. It utilizes the same contrastive representation learning method, MoCo [3], with the exact same loss function, allowing us to derive a comparison between the mathematical behavior of the training pipelines used in both of these studies.

Table 5 demonstrates the comparison in the performance of the two studies. It is important to note that [10] uses a KNN Classifier instead of a full-fledged downstream neural network.

## 6. Discussion

This project originally began as a continuation of previous work with canines [10], motivated by the need to investigate how self-supervised learning methods could reveal latent emotional states in animal species. Shifting our focus from canines to equines was a conscious decision to explore a domain where affective behavior is highly nuanced yet relatively underexamined in the literature. Horses communicate their emotions via a broad spectrum of behavioral cues—facial expressions, ear positions, tail carriage, and subtle body postures—that lend themselves well to data-driven feature extraction. By applying Momentum Contrast (MoCo) [3] to extensive unlabeled equine image data, we aimed to assess whether a CNN encoder could autonomously learn intricate representations that align with distinct emotional or affective states.

We used t-SNE as a visualization tool both before and after pretraining. The result exposed a stark contrast in the way that the network sorts out the feature space. Following training with MoCo, our encoder generated clusters that are significantly more compact and well-delineated than those from the untrained baseline. Our labeled data set (constructed from audio recordings based on the seven canonical Panksepp emotions—CARE, LUST, FEAR, GRIEF, PLAY, RAGE, SEEKING—augmented with PAIN) [9] was relatively modest in size, but applying these features to downstream classification resulted in an accuracy of about 55.47%. A modest performance that underlines the fact that features that are learned in an unsupervised manner do indeed extract relevant characteristics from equine emotional expressions.

One particularly compelling discovery is that the t-SNE plots exhibit more than eight clusters—suggesting the presence of emotional or behavioral nuances beyond our predefined categories. This finding resonates with longstanding proposals in animal behavior research that actual emotional repertoires are more granular than standard frameworks capture [84]. For instance, what we label as “FEAR” might comprise multiple sub-states, each characterized by distinct triggers, intensities, or physiological markers. Thus, our work not only validates the utility of self-supervised learning for uncovering hidden structures in the data but also nudges us toward more fine-grained, ecologically valid categorizations of equine affect [76].

Algorithmic novelty notwithstanding, such an undertaking hints at something much more fundamental: a paradigm shift in animal emotion research, where large-scale data with minimal annotation can create insights that will challenge and sharpen the existing theoretical models. If these new clusters represent new or poorly recognized emotional states, their ability to be consistently identified in many different contexts would call for further empirical study—potentially changing the landscape of how we understand horse welfare, behavior management, and training practices.

### 6.1. Broader Impact on the Field

The broader implications of this study extend well beyond equine research. Our results emphasize the applicability of advanced machine learning and sensor data analysis for unveiling intricate facets of animal behavior and welfare. Traditional methods for studying animal affect typically rely on expert observation and subjective coding schemes, which, although valuable, can be time-consuming and prone to human bias. Self-supervised approaches address these drawbacks by scaling to large datasets without explicit labeling and by allowing the algorithm to surface latent feature representations that might not be evident through manual inspection.

As the cost of sensor and imaging technologies decreases, the potential to extend similar pipelines to other domesticated and even wild species becomes more tangible. The principles demonstrated here, contrastive learning on unlabeled images followed by clustering and minimal-labeled fine-tuning, can also provide information for wider initiatives on animal conservation, zoo management, and public policy on the welfare of animals. More systematically, pain or stress could be detected specifically through these means to facilitate timely intervention, improving treatment results in veterinary medicine.

Crucially, our findings also ignite discussion around the interpretability of “black box” neural networks in sensitive domains such as welfare assessment and medical diagnostics. We must contemplate how best to explain or justify the classification decisions made by these models, especially if their outputs influence real-world decisions regarding an animal’s well-being. Advancements in explainable AI (XAI), tailored to animal-centric applications, could greatly bolster practitioner and stakeholder trust in such systems, enabling ethically responsible deployment [85].

### 6.2. Further Scope of This Research

The encouraging results obtained here naturally suggest several future directions. One line of work could involve capturing a wider range of equine expressions under different environmental conditions, from calm or routine daily scenarios to highly stimulating or stressful events. A richer dataset might yield even finer partitions in the feature space, potentially uncovering sub-emotions or context-specific behaviors that remain occluded in smaller samples. Including temporal information—e.g., video data—would allow us to investigate transitions between emotional states, giving a more dynamic and realistic view of how emotions unfold over time. Three-dimensional CNNs like ResNet-3D-18 can potentially utilize the temporal information from the video data.

Comparisons among different self-supervised learning frameworks, such as SimCLR [2], SwAV [86], or BYOL [4], could also be highly instructive, while MoCo [3] has shown strong performance on many tasks; the choice of architecture and contrastive mechanism can profoundly affect the embeddings extracted. Determining which framework best captures the subtle morphological and behavioral cues of horses could lead to domain-specific methodological improvements, potentially enhancing the recognition rates for more elusive emotional states.

In particular, integrating multimodal data stands out as a promising endeavor. Combining image-based features with audio signals (e.g., whinnies, neighs, snorts), physiological data (heart rate, cortisol levels, respiratory patterns), and contextual metadata (temperature, time of day, type of activity) could illuminate additional facets of equine affect. Such a multi-sensor approach would likely increase confidence in the discovered clusters and their interpretability, reducing the chances of conflating unrelated behaviors with specific emotions.

Finally, there is enormous value in collaborative, cross-disciplinary efforts. Incorporating the expertise of veterinarians, equine behaviorists, animal welfare scientists, and machine learning researchers could foster a mutual exchange of knowledge. Expert insights would refine the labeling of behaviors or emotional states, while robust machine learning techniques would accelerate hypothesis testing and discovery of unrecognized patterns. Ideally, such partnerships would produce validated, high-confidence datasets that better approximate real-world horse-human interactions, offering benefits for sport, therapy, and day-to-day caregiving practices.

### 6.3. Ethical Considerations

Equine emotions trigger a special set of ethical responsibilities. First, it has to be ensured that data collection methods, from taking photographs in private stables to accessing publicly available imagery, are strictly adhering to guidelines protecting animal welfare. Although we work with our own dataset of wild horse images, future expansions may include more invasive modalities, such as biometric sensors or close-up videography, which would require careful ethical review to ensure that the animals are not unduly stressed.

Second, the interpretation of discovered clusters demands caution. Although the model may autonomously group visually similar behaviors, these clusters are not automatically “new emotions”. Overinterpretation risks anthropomorphizing or mislabeling a horse’s response, potentially leading to misguided interventions or welfare policies. As such, the labeling process must remain tightly coupled with domain expertise to validate whether observed behaviors are consistent with ethologically recognized expressions (e.g., ear-pinning for agitation, lip-lifting for pain or discomfort).

Any practical application of these tools must be done with an appreciation for how classification errors might translate into real-world consequences. Erroneous classification (for example, labeling a tired horse as depressed or a nervous horse as aggressive) may result in inappropriate management decisions. This is where transparency and explainability become very important: stakeholders must understand that these models provide probabilistic suggestions, not definitive diagnoses. Frameworks for risk assessment and error mitigation should be put in place, especially in cases where an incorrect label could materially affect an animal’s treatment or living conditions.

Lastly, data privacy and ownership considerations cannot be overlooked, even in animal-focused research, while horses do not have the same legal protections as humans regarding “personal data,” the concerns of owners, caretakers, and property managers must be respected. Clear data governance policies—covering consent for image capture, usage rights, and the secure handling of proprietary biometric information—are foundational to ethical research practices. By addressing these concerns proactively, we can ensure that the pursuit of scientific discovery does not come at the expense of animal welfare or stakeholder trust.

## 7. Conclusions

In summary, this study shows that self-supervised learning, specifically Momentum Contrast (MoCo), is capable of revealing latent structures in equine emotional behavior from large-scale, unlabeled image datasets. Our approach was able to extract meaningful feature representations, which enhanced downstream classification tasks and opened the possibility for states of emotions beyond those usually categorized. The t-SNE visualizations, indicating the presence of further clusters, indicate that equine emotional repertoires could be more fine-grained than the current conceptualizations acknowledge, and the current study makes a prima facie case for additional empirical work. This work not only demonstrates the utility of data-driven approaches in driving animal emotion research but also underlines the need for interdisciplinary approaches to refine our understanding of animal welfare and behavior. Future studies will be able to extend from these findings to consider nuances in specific contexts and to include multimodal data to better understand the emotional lives of animals. Please visit the Appendix A to view a detailed mapping of discussed concepts.

## Figures and Tables

**Figure 1 sensors-25-00859-f001:**
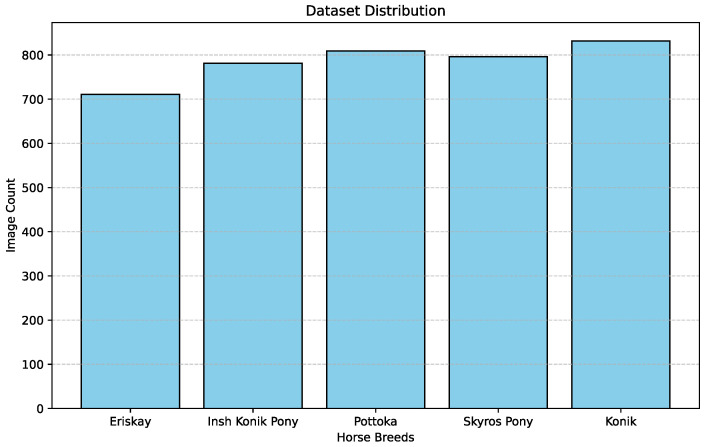
Distribution of the image count of each individual horse breed within our dataset.

**Figure 2 sensors-25-00859-f002:**
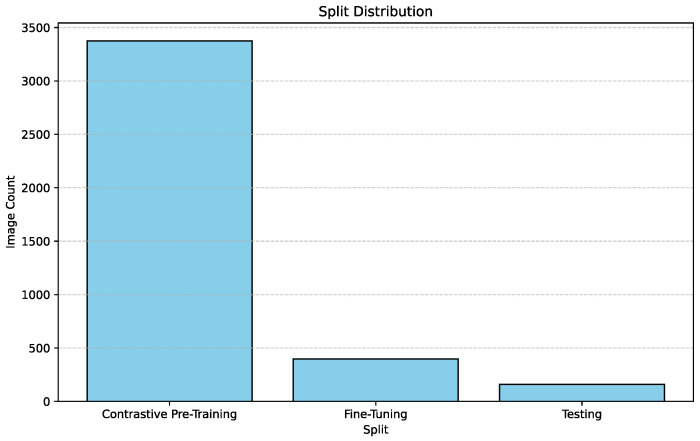
Distribution of the image count in each split.

**Figure 3 sensors-25-00859-f003:**
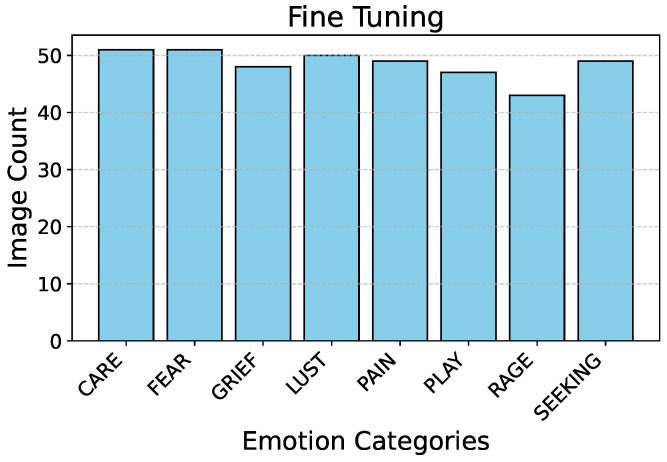
Distribution of emotions in fine-tuning split.

**Figure 4 sensors-25-00859-f004:**
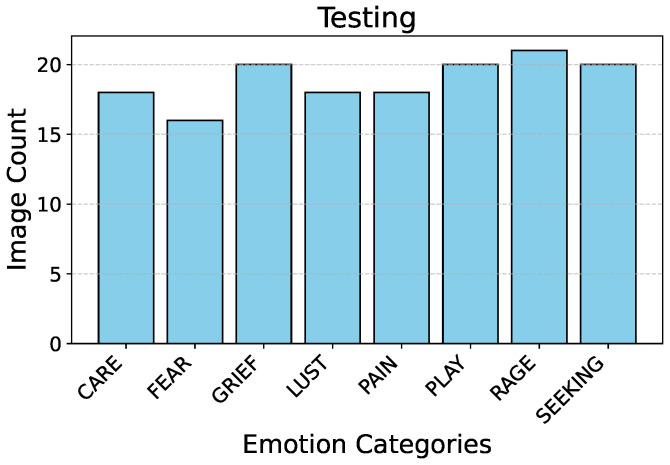
Distribution of emotions in test split.

**Figure 5 sensors-25-00859-f005:**
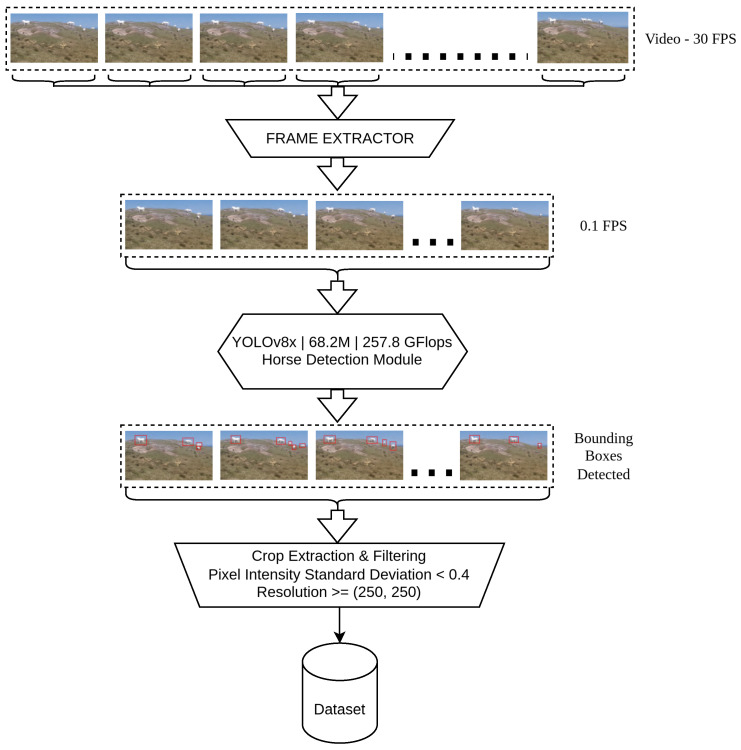
Image retrieval pipeline.

**Figure 6 sensors-25-00859-f006:**
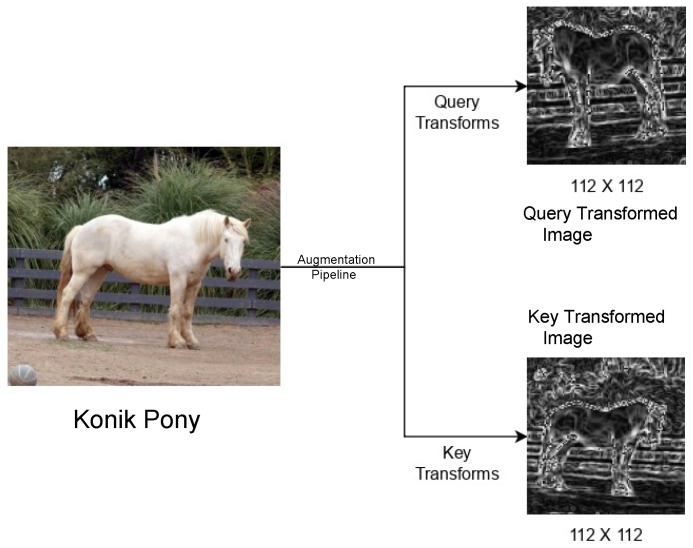
An Eriskay Pony image before and after augmentation.

**Figure 7 sensors-25-00859-f007:**
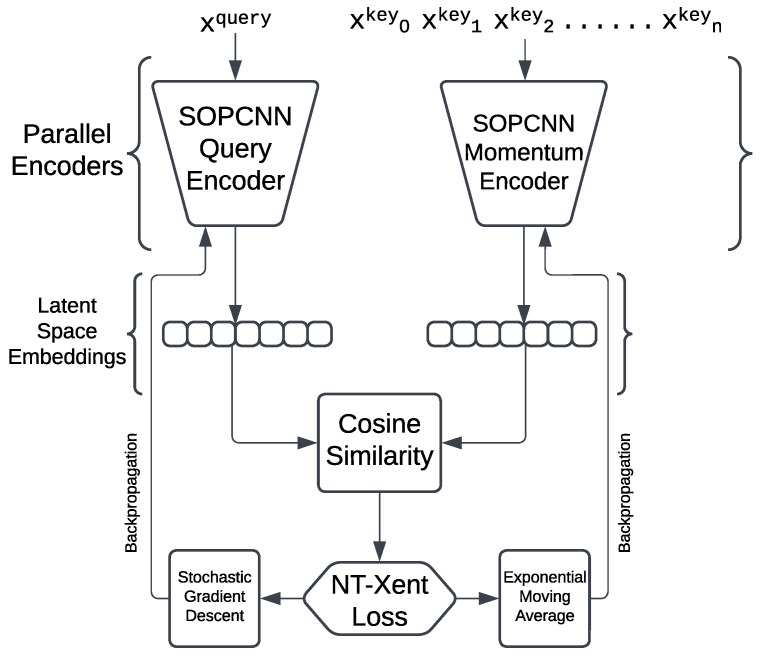
Final MoCo network architecture for contrastive pre-training.

**Figure 8 sensors-25-00859-f008:**
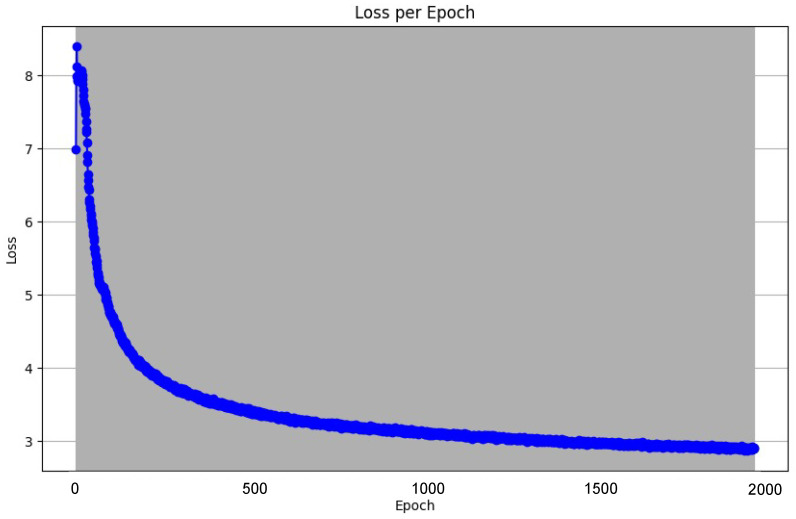
Pre-training loss.

**Figure 9 sensors-25-00859-f009:**
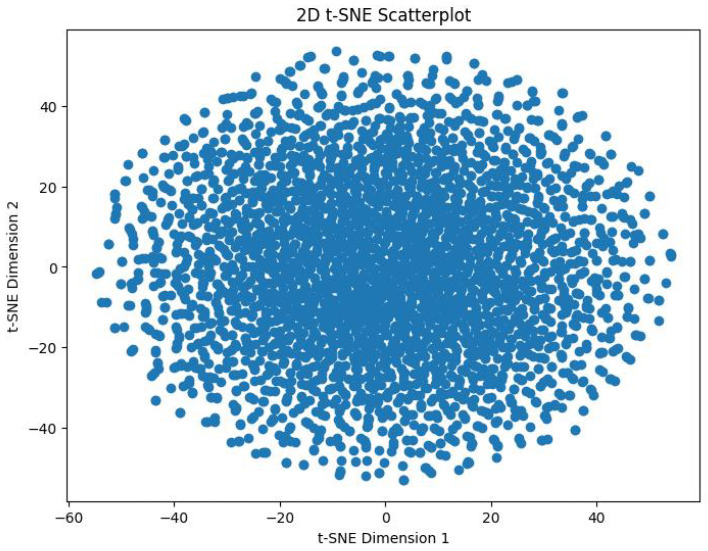
Before Training.

**Figure 10 sensors-25-00859-f010:**
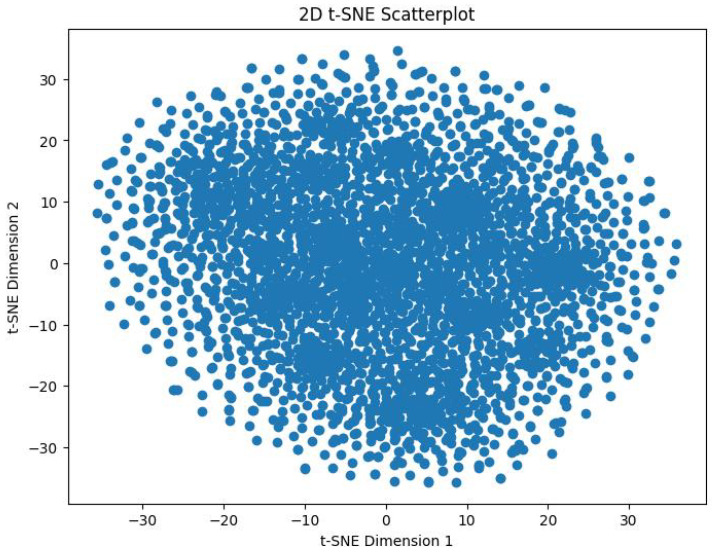
After training.

**Figure 11 sensors-25-00859-f011:**
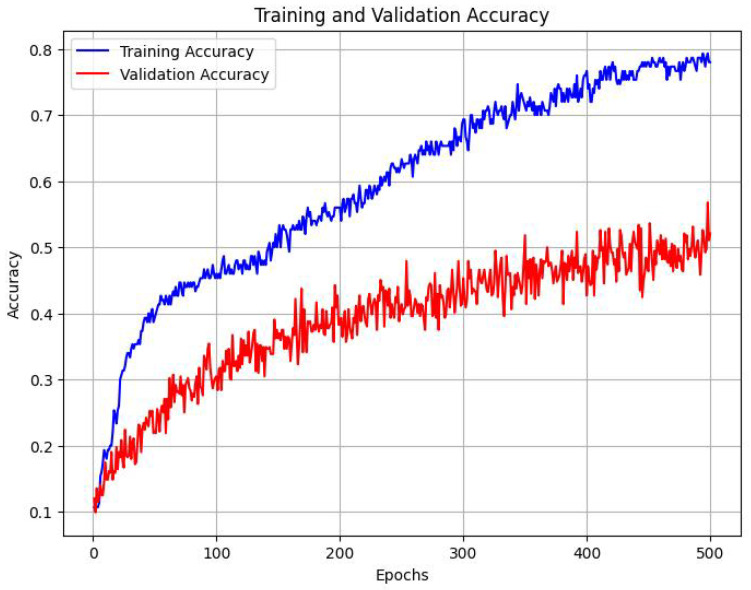
Training and validation accuracy.

**Table 1 sensors-25-00859-t001:** Description of wild horse breeds.

Horse Breed	Location	Images
Eriskay Pony	Eriskay, Western Scotland	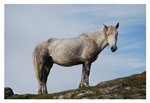
Insh Konik Pony	RSPB Insh Marshes, Scotland	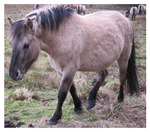
Pottoka	Piornal, Spain	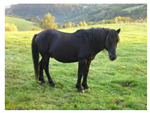
Skyros Pony	Skyros, Greece	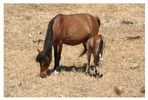
Konik	Wicken Fen Nature Reserve, England	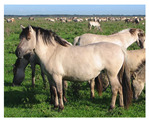

**Table 2 sensors-25-00859-t002:** Hardware specifications used for the deep learning project.

Component	Specifications
GPU	NVIDIA Tesla P100
GPU Memory	16 GB HBM2
Compute Units	3584 CUDA Cores
CPU	Intel Xeon (Kaggle)
RAM	13 GB
Platform	Kaggle Kernel

**Table 3 sensors-25-00859-t003:** Hyperparameters used for pretraining the CNN using momentum contrast (MoCo).

Hyperparameter	Value	Notes
Input Size	112	Dimensions of the input image in pixels
Batch Size	32	Number of training samples processed at once
Momentum Coefficient	0.999	Coefficient for the momentum in the MoCo algorithm
Temperature Parameter	0.07	Scaling factor for the contrastive loss function
Queue Capacity	65,536	Total number of negative keys stored in the queue
Training Epochs	2000	Number of complete passes through the dataset
Representation Dimension	64	Dimensionality of the output feature representations
Weight Decay	$1\times 10^{-4}$	Regularization term to reduce overfitting
Learning Rate	0.0001	Step size used in the optimization process
Optimizer Momentum	0.9	Momentum parameter used in the optimizer
Perplexity	30	Perplexity parameter for t-SNE visualization

**Table 4 sensors-25-00859-t004:** Emotion labels and their corresponding accuracy scores.

Emotion	Accuracy
Care	59.50%
Fear	55.40%
Grief	49.30%
Lust	65.56%
Pain	57.60%
Play	52.10%
Rage	62.20%
Seeking	42.10%

**Table 5 sensors-25-00859-t005:** Comparison in Emotion labels and their corresponding accuracy scores.

Emotion	Results in Horses	Results from [10] in Dogs
Care	59.50%	34.61%
Fear	55.40%	35.51%
Grief	49.30%	59.50%
Lust	65.56%	62.79%
Play	52.10%	38.88%
Rage	62.20%	45.91%
Seeking	42.10%	40.40%
Pain	57.60%	-

## Data Availability

Our source code is available at https://github.com/caffeinekeyboard/Horse_Emotion_Classification (accessed on 22 January 2025).

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
