# Peer review of "Identifying Novel Emotions and Wellbeing of Horses from Videos Through Unsupervised Learning"

_sensors, 2025, doi:10.3390/s25030859_

Round 1
Reviewer 1 Report
Comments and Suggestions for Authors
The paper discusses an approach to identifying horse emotions using unsupervised learning techniques. The authors construct a high-quality, diverse dataset of horse images and apply the Momentum Contrast (MoCo) framework to predict emotions based on the seven Panksepp emotions and an additional "Pain" emotion. The study aims to improve the understanding of equine emotions and enhance human-animal communication.
Here are some questions and/or comments regarding the paper.
1. How did you ensure the diversity and representativeness of the dataset across different horse breeds and geographical locations?
2. Were there any specific challenges encountered while collecting and processing the video footage?
3. Can you elaborate on the choice of the SOPCNN encoder over traditional encoders like ResNet-50? What specific advantages did it offer?
4. How did you determine the optimal hyperparameters for training the encoder using the MoCo framework?
5. The downstream classifier achieved a classification accuracy of 55.47%. Can you elaborate more on similar tasks to other state-of-the-art models?
6. Can you provide more details on the individual emotion accuracies? What factors might have contributed to the varying accuracies across different emotions?
Reviewer 2 Report
Comments and Suggestions for Authors
General comments:
This paper focuses on the novel and challenging topic of using unsupervised learning techniques to identify the emotions of horses, which has certain innovation and application value. The author has taken many effective measures, such as constructing a new dataset, improving the contrastive learning framework and designing a customized downstream classifier, etc., thus achieving meaningful results in the field of horse emotion recognition and opening up new directions for the fields of animal-computer interaction and animal emotion computing. However, there are also some aspects in the paper that need improvement.
From the perspective of the model's performance, although it can identify the emotions of horses to a certain extent, there is still much room for improvement in the overall classification accuracy. For example, the accuracy of the downstream classifier on 8 labels is only 55.47%, and the identification of some emotions is not precise enough. In terms of data utilization, the experiment only relies on image data and does not fully utilize the temporal information in video data, which makes it difficult to fully present the dynamic change process of horses' emotions.
Furthermore, in terms of comparative analysis, the paper mainly compares with traditional handcrafted feature methods, but lacks a detailed comparison with other advanced unsupervised and semi-supervised learning methods, failing to fully demonstrate its advantages. Moreover, the interpretability of the neural network model in the paper is not strong, and there is a lack of in-depth analysis of how it learns and distinguishes the emotions of horses, which may affect readers' trust in the results in practical applications.
Specific Review Comments:
1. Add comparative experiments with other relevant advanced methods, and conduct a detailed analysis of the advantages and disadvantages of the proposed method in terms of different metrics to further verify its effectiveness and innovativeness. In the comparative experiments, ensure the fairness and rationality of the experimental settings, including aspects such as the division of the dataset and the adjustment of hyperparameters.
2. Suggestions regarding the chart presentation in the paper. Data presentation is relatively intuitive. For example, using multiple bar charts and scatter plots to show data and training situations is helpful for understanding the experimental setup. However, the deficiencies are also obvious. Firstly, the labeling of the axes, legends, etc. of some charts is not detailed enough. For example, Figure 6 should label the image enhancement-related information in more detail. Secondly, the types of charts are relatively single, and more diverse charts are needed to assist understanding when dealing with complex content.
3. Regarding the issue of presenting specific code in the algorithm part of the paper. The algorithms in the paper are presented in the form of code fragments instead of pseudo-code-like algorithms. The actual code causes difficulties in understanding the algorithms. Although it is beneficial for result reproduction and application, it reduces readability and universality and easily distracts attention from the core of the algorithm.
4. In terms of the number of formulas, the paper mainly involves the NT-Xent loss function formula in contrastive learning, which is used to train and optimize the model. Compared with some research papers with extremely strong theoretical nature, the number of formulas is relatively small. Adding formula derivations to this paper usually can enhance the theoretical depth. For example, in the parts of image enhancement and downstream classifier design, if there are formula derivations showing their impact on feature distribution, classification results and the internal mathematical relationships, it can enable readers to clearly understand the theoretical process from input to output, making it not just present the experimental results.
5. Adopt interpretability techniques, such as visualization techniques, to analyze and explain the decision-making process of the model, reveal how the model recognizes different emotions based on the image features of horses, improve the transparency and interpretability of the model, and make the research results more persuasive and credible.
Round 2
Reviewer 2 Report
Comments and Suggestions for Authors
Dear Editor and Authors,
Thank you for the detailed response and revisions to the review comments. After careful reading and consideration, I believe the authors have adequately addressed the reviewers' comments and made corresponding modifications to the paper. Below is a response to the main comments:
1. Regarding the improvement of relying solely on image data for experiments: The authors discussed the potential use of 3-Dimensional CNNs in Section 6.2 to fully utilize the temporal information in videos.
2. Regarding the lack of detailed comparison with other semi-supervised learning methods: The authors have compared several aspects of these studies. References [10-14], [73], and [74] have been added.
3. Regarding the lack of in-depth analysis of how the model learns and distinguishes horse emotions: The authors have provided additional explanations in Section 4.3.1. Additionally, t-SNE visualizations were used to analyze the clustering of learned embeddings, clearly illustrating how the model autonomously distinguishes between emotion categories.
4. Regarding the formatting issues in Figure 6: The authors have modified Figure 6 to add more details and improve clarity for readers.
5. Regarding the insufficient theoretical explanation of the formula: The authors have added an intuitive derivation of the NT-Xent Loss function in lines 602-612 after Figure 7.
Overall, the authors have made careful revisions to the language and formatting, significantly improving the overall quality of the paper. I believe the authors have adequately addressed the review comments and made the necessary modifications to the paper. Therefore, I recommend accepting this paper.